# Seeking Evidence of The MAGA Cult and Trump Derangement Syndrome: An Examination of (A)symmetric Political Bias

Andrew S. Franks [1,2,*] and Farhang Hesami [3]

1 Psychology, Liberal Arts and Social Sciences, Researcher, Central Michigan University, Mount Pleasant, MI 48859, USA
2 Social, Behavioral and Human Sciences, Interdisciplinary Arts and Sciences, Faculty, University of Washington Tacoma, Tacoma, WA 98402, USA
3 Psychology, Social Sciences, Alumnus, Pacific Lutheran University, Tacoma, WA 98447, USA; farhang.hesami@gmail.com
* Correspondence: andrew.franks@cmich.edu

**Abstract:** Three studies sought to explore the existence of (a)symmetric bias regarding Donald Trump. In Study 1, participants read one of three statements expressing different degrees of favorability toward electing the President of the United States via a National Popular Vote attributed to Trump or an anonymous source. In Study 2, participants read one of two statements either favoring or disfavoring the name change of the Washington NFL franchise, and the statement was attributed to either Trump or an anonymous source. In Study 3, Trump and Biden voters were asked to rate their support or opposition to counting all the votes in battleground states when continued counting was expected to either help Trump or Biden. Results for all three studies supported the asymmetric bias hypothesis. Trump supporters consistently showed bias in favor of the interests and ostensible positions of Trump, whereas Trump's detractors did not show an opposing bias.

**Keywords:** motivated social cognition; ideological asymmetry; political bias; political psychology; Donald Trump

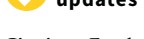



## 1. Introduction

*"It's no longer a political party. It's a cult."* -Former Republican Representative Mickey Edwards on leaving the Republican Party (January 2021).

The United States is experiencing an era of extreme political polarization, which may be on a nigh-unstoppable upward trajectory [1]. Political leaders themselves are a particularly strong motivating force of this trend [2], and there may be no single individual more responsible for polarizing the American people over the past half-decade than the 45th President of the United States, Donald J. Trump. For instance, Trump has repeatedly attempted to shift blame for the nation's poor outcomes during the coronavirus pandemic from his own administration to political rivals in "Blue States" [3], and he has hindered such states' efforts to obtain much-needed medical equipment to treat COVID-19 [4]. Along with his Attorney General Bill Barr, Trump designated several cities run by Democratic mayors as "anarchist jurisdictions" and threatened to cut off their federal funding [5]. Furthermore, Trump has repeatedly failed to condemn white supremacist and nationalist groups, likely playing a role in emboldening a domestic terrorist plot to kidnap Michigan Governor Gretchen [6]—or as Trump calls her "that woman from Michigan [7]—and an attempted insurrection at the U.S. Capitol in Washington, D.C. It should be no surprise then that research has directly implicated Trump in both causing and benefitting from polarization related to sexism, racial resentment, partisanship, and polarization [8–11]. As such, the current research is meant to examine the symmetry or asymmetry in the degree to which Trump's rhetoric and interests affect the political opinions of his supporters and detractors.

### 1.1. On "Cults" and "Derangement"

Trump's followers are often derided as a cult [12], blindly accepting the words of their authoritarian leader as gospel, whereas his detractors are often accused of suffering from "Trump Derangement Syndrome," a supposed mental disorder characterized by an irrational hatred for the man and his policies [13]. To what degree are these dueling accusations true? Is it possible that Trump's rhetoric is uncritically accepted by his supporters and reflexively rejected by his detractors? The current research sought to examine the degree to which Trump's supporters and detractors are influenced to support positions that would typically be considered congenial and uncongenial to their political orientation as conservatives or liberals. In order to do so, we need to consider two alternative effects: (1) a symmetric bias effect whereby Trump's supporters and detractors show an equal propensity for supporting or opposing his positions and interests; and (2) an asymmetric bias effect whereby participants on one side (his supporters) are heavily biased in favor of his messages and interests, and participants on the other side (his detractors) are less affected or largely unbiased in opposition to his messages and interests.

### 1.2. Evidence for Asymmetric Political Bias

Meta-analyses demonstrating asymmetrical levels of bias between liberals and conservatives have found greater degrees of politically motivated bias among conservatives [14]. Conservatives have been shown to be more likely to attribute scientific findings that are discordant with their political beliefs to liberal bias in the researchers (compared to liberals attributing discordant findings to conservative bias in the researchers) [15]. Using data collected over several decades from large, nationally representative data sets such as the American National Election Survey and General Social Survey, Morisi and colleagues [16] found that American liberals were more likely to trust and accept the legitimacy of the government during Republican presidential administrations than American conservatives were to trust and accept the legitimacy of the government during Democratic presidential administrations (see also "Stop the Steal"). Conservatives and liberals have also been shown to differ in the degree of partisanship in their preferred media sources and in their likelihood of spreading political misinformation [17], and although both liberal and conservative media have been found to contribute to polarization, the strength of such effects seems to be stronger on the right [18]. Numerous meta-analyses by Jost and colleagues [14,19,20] have detailed widespread research findings demonstrating asymmetries in myriad political biases, personality characteristics, cognitive abilities and preferences, and neurocognitive characteristics. For instance, conservatives have greater degrees of belief in a just world [21] and social dominance orientation [22,23]. Liberals are higher in openness to experience, and conservatives are higher in conscientiousness [24]. Liberalism is associated with a preference to engage in deliberative, analytical thinking [25,26] and higher scores on tests of cognitive ability [22] as well as cognitive flexibility [27]. Neurocognitive differences between liberals and conservatives have been found in brain areas such as the anterior cingulate cortex [28,29]—an area associated with, among other things, the ability to tolerate ambiguity—and amygdala [29]—an area associated with, among other things, negative emotional responses.

Given this range of behavioral, attitudinal, personality, cognitive, and neurocognitive asymmetries, it is justified to hypothesize asymmetrical reactions to messages attributed to Donald Trump.

**Hypothesis 1 (H1).** *Asymmetric Trump bias hypothesis. Across all studies, Trump supporters will show increased bias favoring his ostensible messages and interests, whereas Trump detractors will not show symmetric bias in opposition to his ostensible messages and interests.*

### 1.3. Evidence for Symmetric Political Bias

Meta-analyses have also been conducted on research purported to demonstrate symmetrical political bias [30], although the conclusions of such meta-analyses have been

directly challenged [31]. Research on right-wing authoritarianism [32] has produced hundreds or even thousands of peer-viewed research articles over the past four decades, but more recent research has suggested the simultaneous existence of "left-wing authoritarianism" [33]. Conway and colleagues [34] also called into question ideological differences in cognitive rigidity, whereas additional research has found conservatives and liberals to be equally prejudiced against each other and ideologically dissimilar groups [35–37] and equally motivated to avoid each other's opinions [38]. Liberals and conservatives have also been shown to be more skeptical of research findings that are politically incongruent to their own beliefs [39] and to support/oppose policies based on the party that ostensibly proposed them regardless of the content of the policy (the "party over policy" effect) [40].

This last finding is particularly germane to the current research; thus, a competing hypothesis must be advanced.

**Hypothesis 2 (H2).** *Symmetric Trump bias hypothesis. Across all studies, Trump supporters will show increased bias favoring his ostensible messages and interests, and Trump detractors will also show symmetric bias in opposition to his ostensible messages and interests.*

*1.4. Overview*

The current research sought to examine whether Donald Trump promotes political polarization symmetrically by shifting his supporters' views toward whatever position he advocates and his detractors' views against whatever position he advocates. Across Studies 1 and 2, statements in favor of various positions on current sociopolitical issues were attributed to either Donald Trump or to an anonymous source. Topics were selected in order to plausibly attribute differing opinions to Trump. In Study 1, statements were crafted to communicate differing levels of support for a National Popular Vote—something that conservatives currently tend to oppose and liberals support (two Republicans have assumed the presidency despite losing the popular vote over the last twenty years), although Trump once claimed to support it during the 2016 election cycle. In Study 2, statements were crafted to communicate differing levels of support for the recent name change of the former "Washington Redskins" to the "Washington Football Team" on purely financial grounds. The focus on the financial aspect allowed for both opposing positions to be plausibly attributed to Trump, while also testing whether attitudinal change regarding the name change financially speaking would carry over to attitudes regarding the moral correctness of the name change. Finally, a third study was conducted in the aftermath of the 2020 U.S. Presidential Election, where votes were counted in key battleground states for days after the election leading to Joe Biden winning several states where Trump appeared to be "leading" on election night. This additional study allowed for a more concrete operationalization of "Trump supporters" (i.e., those who had voted for him that same week) and to examine whether liberals were biased toward the interests of the Democratic nominee (Joe Biden). Study materials, hypotheses, and planned analyses were preregistered and can be accessed in an open science forum (https://osf.io/wt635/ accessed on 3 September 2021).

**2. Methods**

*2.1. Participants*

Using Amazon's Mechanical Turk (mTurk) system and limiting enrollment in the study to "Master Workers" in the United States, 394 participants ($M_{age}$ = 34.95, SD = 13.56) were recruited and paid USD 0.50 for completing the study. G*Power 3.1 [41] recommended a sample size of 351 to find small effects (f = 0.15) with power = 0.80 at conventional levels of significance ($p < 0.05$) for the planned analysis of variance. Accordingly, data collection was halted shortly afterward, with some additional leeway for the possibility of some participants failing attention checks. The sample was 43% female, 73% non-Hispanic white, had a median education of a bachelor's degree, and a median income between USD 25,000 and USD 49,999. The sample was evenly divided between being some denomination of Christian (47%) and being non-religious (47%).

*2.2. Materials and Procedure*

After agreeing to complete the survey on mTurk, participants followed a link to a survey hosted on Qualtrics where they completed an online consent document. Participants read one of three statements on the prospect of the President of the United State being elected via a National Popular Vote (NPV). One statement was anti-NPV, one was neutral on NPV, and one was pro-NPV (see Appendix A). The statements were attributed to either Donald Trump or an anonymous "political pundit." As the dependent measure, participants indicated their level of agreement with the statement they read on 7-point scales, with high values indicating more agreement. Afterwards, participants answered two attention/manipulation check items asking them to identify the basic content of what they read and to whom the statements were attributed. All participants passed both checks, as largely expected from Master Workers. Participants also indicated the likelihood that they would vote for Trump in the 2020 election on a 7-point scale, with 1 indicating "No chance" and 7 indicating that voting for Trump was a certainty and reported their general political orientation on a 1 "Very Liberal" to 7 "Very Conservative" scale. Finally, participants completed a demographic questionnaire, received a completion code, and returned to mTurk to enter the code.

## 3. Results

*3.1. Planned Analyses*

For the Trump support variable, participants were categorized as anti-Trump if they indicated that there was "No chance" that they would vote for Trump in 2020 and pro-Trump if they indicated any non-zero likelihood of voting for Trump in 2020. This classification was performed to create equal groups (because about half of participants indicated that there was no chance of them voting for Trump) and also in order to provide the strongest chance of finding symmetrical bias.

A 2 (Trump support: Trump detractors vs. Trump supporters) $\times$ 2 (source: anonymous vs. Trump) $\times$ 3 (position: anti-NPV vs. neutral-NPV vs. pro-NPV) analysis of variance was conducted, with the level of agreement with the statement as the dependent variable. The main effect of Trump support was significant, indicating overall higher levels of agreement from Trump supporters (M = 4.18, SD = 1.84) than Trump detractors (M = 3.46, SD = 1.82), $F(1, 382) = 15.31$, $\eta p^2 = 0.039$, $p < 0.001$. The main effect of source was significant as well, indicating that statements elicited more agreement when attributed to Trump (M = 3.92, SD = 1.89) than to an anonymous political pundit (M = 3.64, SD = 1.83), $F(1, 382) = 4.37$, $\eta p^2 = 0.011$, $p = 0.037$. The main effect of position was non-significant, $F(2, 382) = 1.22$, $\eta p^2 = 0.006$, $p = 0.298$.

Both significant main effects were qualified by significant two-way interaction effects. The Trump Support $\times$ Source interaction was significant, $F(1, 382) = 4.91$, $\eta p^2 = 0.013$, $p = 0.027$. Probing this interaction revealed that Trump supporters were more likely to agree with statements attributed to Donald Trump (M = 4.64, SD = 1.80) rather than to an anonymous political pundit (M = 3.82, SD = 1.81), 95% $CI_{Diff}$ [−1.32, −0.31], $p = 0.007$, whereas Trump detractors were not significantly affected by the source of statements, $M_{anon} = 3.42$, $SD_{anon} = 1.84$, $M_{Trump} = 3.48$, $SD_{Trump} = 1.80$, 95% $CI_{Diff}$ [−0.53, 0.39], $p = 0.663$. This pattern is consistent with Hypothesis 1 (Asymmetric Trump Bias) and inconsistent with Hypothesis 2 (Symmetric Trump Bias), and is illustrated in Figure 1.

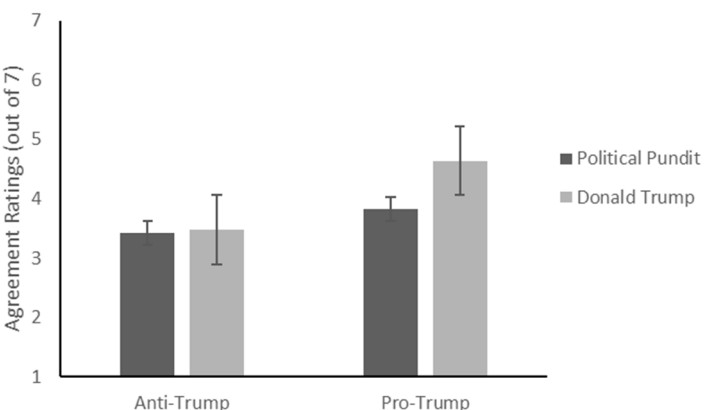

**Figure 1.** Agreement with the statement on a National Popular Vote by the participant with Trump support and a statement source.

The Trump Support × Position interaction was also significant, F(2, 382) = 10.07, $\eta p^2$ = 0.050, *p* < 0.001. Probing this interaction showed that Trump detractors preferred the pro-NPV message (M = 4.18, SD = 1.91) to both of the other statements: anti-NPV (M = 2.87, SD = 1.54), 95% CI_Diff [0.83, 1.94], *p* < 0.001, and neutral (M = 3.36, SD = 1.76), 95% CI_Diff [0.27, 1.41], *p* = 0.004. For Trump detractors, this difference between the anti-NPV and neutral statements was nonsignificant (*p* = 0.06). Trump supporters did not show significantly different levels of agreement based on the content of the statements (ps > 0.47). This pattern of results implies that participants categorized as Trump detractors responded consistently with the typical current attitudes of liberals regarding NPV regardless of the source of the message, which is to say that they seem to be responding to the message itself rather than the source.

### 3.2. Exploratory PROCESS Analysis

An exploratory moderation analysis using PROCESS Model 1 [42] demonstrated that the Source × Trump Support interaction remained significant when Trump support was entered into the model as a continuous variable (rather than as a dichotomous variable as in the ANOVA above): b = 0.22, 95% CI [0.03, 0.40], *p* = 0.022. Probing this interaction demonstrated that the effect of the source manipulation on agreement ratings was significant at the 62nd percentile of Trump support and higher, as illustrated in the Johnson–Neyman plot in Figure 2.

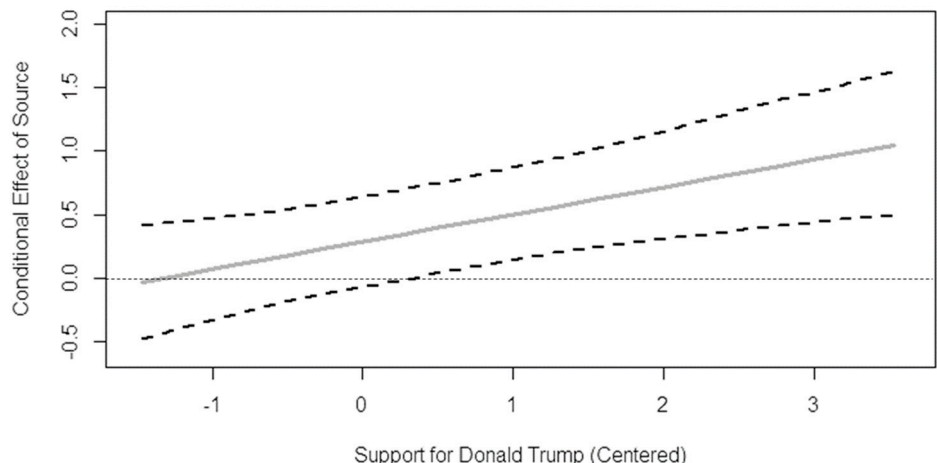

**Figure 2.** Effect of source on agreement ratings by the degree of Trump support with 95% confidence intervals.

In addition, those who are concerned about whether these effects specifically regarding Trump support generalize more broadly to typical liberal/conservative political bias should know that Trump support and self-reported political orientation were very highly correlated, r = 0.762, $p < 0.001$. A parallel exploratory PROCESS Model 1 analysis with political orientation as the moderating variable revealed a significant Source × Political Orientation interaction, b = 0.19, 95% CI [0.00, 0.380], $p = 0.047$. Probing this interaction demonstrated that the effect of the source manipulation on agreement ratings was significant at the 50th percentile of political orientation and higher.

These exploratory analyses lend additional support to the asymmetric bias hypothesis, while establishing that the processes regarding Trump support generalize to political biases more generally.

### 4. Discussion

Study 1 supported the Asymmetric Trump Bias Hypothesis. Trump supporters showed an increased likelihood of agreeing with any position regarding NPV as long as it was attributed to Trump. The source of the message did not affect agreement ratings among Trump detractors, who did not reflexively reject any position advocated by Trump as may be expected if "Trump Derangement Syndrome" was a thing. Exploratory analyses also demonstrated that the higher the level of Trump support, the greater the level of bias. This was also the case with higher levels of conservatism. When it comes to Trump, it seems that his supporters and those high in conservatism generally experience motivated social cognition to adopt his position on issues.

### 5. Study 2

Study 2 sought to replicate the asymmetric bias effect observed in Study 1 with a much different sociopolitical issue: the recent name change of the NFL's Washington Football Team. This issue has some important differences from the NPV issue from Study 1 that would, if results are consistent, provide strong evidence that asymmetric Trump bias influences Trump supporters' opinions on a wide array of topics. The name change of the Washington Football team is one such issue, with political disagreement even about the correct reasons for doing so [43]. As recently as July 2020, only 29% of American adults thought it was necessary to change the name, whereas 49% outright opposed changing it [44]. Accordingly, there should be enough room for the attitudes of Trump's detractors to move in either direction away from Trump's ostensible position in this study. In addition, this issue is one on which support for the change could hinge on both financial concerns (e.g., it would be profitable/unprofitable to change the name) and moral concerns (e.g., the name needs to change because it is offensive). This allowed for boundary conditions to be explored regarding biased responses to Trump as a source of information. Statements in this study are made in the context of the financial concerns of Washington Football Team's name change, but agreement with the name change was measured both in terms of financial and moral aspects. Consequently, we can examine if Trump's influence on a particular issue is narrowly constrained to the explicit contents of his statements or carries over into other appraisals of target issues.

### 6. Methods

#### 6.1. Participants

Using Amazon's Mechanical Turk (mTurk) system and limiting enrollment in the study to "Master Workers" in the United States, 251 participants ($M_{age}$ = 35.53, SD = 13.46) were recruited and paid USD 0.50 for completing the study. G*Power 3.1 [41] recommended a sample size of 199 to find small effects (f = 0.15) with power = 0.80 at conventional levels of significance ($p < 0.05$) for the planned analysis of variance. Accordingly, data collection was halted shortly after, providing leeway for the possibility of data exclusions for failing attention checks. The sample was 41% female, 72% non-Hispanic white, had a median education of a bachelor's degree, and a median income between USD 50,000 and

USD 74,999. The sample was about evenly divided between being some denomination of Christian (49%) and being non-religious (46%).

*6.2. Materials and Procedure*

After agreeing to complete the survey on mTurk, participants followed a link to a survey hosted on Qualtrics where they completed an online consent document. Participants read one of two statements on the recent name change of the Washington NFL franchise from "Washington Redskins" to "Washington Football Team". One statement disparaged the change on financial grounds, whereas the other statement praised the change on financial grounds (see Appendix B). The statements were attributed to either Donald Trump or an anonymous "sports broadcaster." As the dependent measures, participants indicated their level of support for the recent name change on financial and moral grounds on two separate 7-point scales, with high values indicating more support for the change. Afterwards, participants answered two attention/manipulation check items asking them to identify the basic content of what they read and to whom the statements were attributed. All participants passed both checks. Participants also indicated the likelihood that they would vote for Trump in the 2020 election on a 7-point scale, with 1 indicating "No chance" and 7 indicating that voting for Trump was a certainty; they also reported their general political orientation on a 1 "Very Liberal" to 7 "Very Conservative" scale. Finally, participants completed a demographic questionnaire, received a completion code, and returned to mTurk to enter the code.

**7. Results**

*7.1. Planned Analyses*

For the Trump support variable, participants were categorized as anti-Trump if they indicated that there was "No chance" that they would vote for Trump in 2020, and pro-Trump if they indicated any non-zero likelihood of voting for Trump in 2020. This classification was performed to create equal groups (because about half of participants indicated that there was no chance of them voting for Trump) and also in order to provide the strongest test of symmetrical bias.

A pair of 2 (Trump support: Trump detractors vs. Trump supporters) × 2 (source: anonymous vs. Trump) × 2 (position: anti-name change vs. pro-name change) analyses of variance was conducted, with the level of agreement with the name change on each of the following as dependent variables: (1) financial grounds and (2) moral grounds.

7.1.1. Financial Agreement

The main effect of Trump support was significant, indicating overall higher levels of agreement from Trump supporters ($M = 4.18$, $SD = 1.84$) than Trump detractors ($M = 3.46$, $SD = 1.82$), $F(1, 382) = 15.31$, $\eta p^2 = 0.039$, $p < 0.001$. There were no significant main effects of any individual variable on Financial Agreement, $Fs(1, 243) < 3.52$, $ps > 0.06$. The three-way interaction was also non-significant.

Both two-way interaction effects were significant. First, the Trump Support × Source interaction was significant, $F(1, 243) = 5.45$, $\eta p^2 = 0.022$, $p = 0.020$. Probing this interaction revealed that Trump supporters were more likely to agree with the name change on financial grounds when an argument in favor of it was attributed to Donald Trump ($M = 4.92$, $SD = 1.61$) rather than to an anonymous political pundit ($M = 4.16$, $SD = 1.58$), 95% $CI_{Diff}$ [−1.41, −0.11], $p = 0.026$, whereas Trump detractors were not significantly affected by the source of statements, $M_{anon} = 4.27$, $SD_{anon} = 1.82$, $M_{Trump} = 3.99$, $SD_{Trump} = 1.75$, 95% $CI_{Diff}$ [−0.31, 0.81], $p = 0.410$. This pattern is consistent with Hypothesis 1 (Asymmetric Trump Bias) and inconsistent with Hypothesis 2 (Symmetric Trump Bias), and is illustrated in Figure 3.

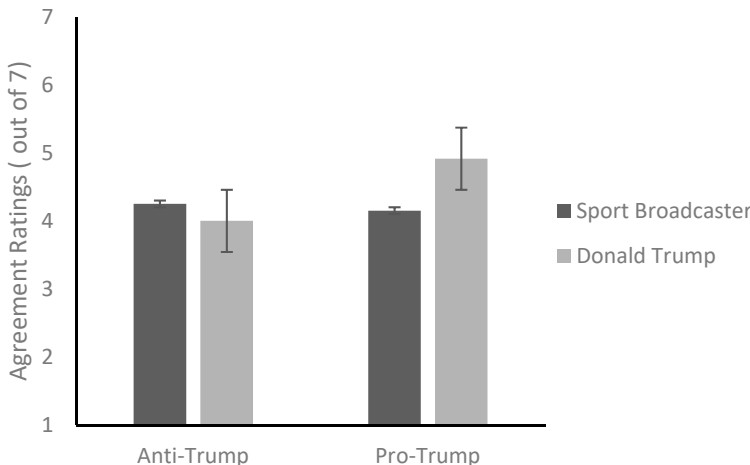

**Figure 3.** Agreement with the team name change on financial grounds by participant on the degree of Trump support and statement source.

The Trump Support × Position interaction was also significant, $F(1, 243) = 6.25$, $\eta p_2 = 0.025$, $p = 0.013$. Probing this interaction showed that Trump detractors agreed more strongly with the name change on financial grounds when reading a message supporting it ($M = 4.44$, $SD = 1.91$) than when reading a message opposing it ($M = 3.83$, $SD = 1.54$), 95% $CI_{Diff}$ [$-1.17$, $-0.06$]. Trump supporters did not show significantly different levels of agreement based on the content of the statements ($p = 0.15$). This pattern of results implies that participants categorized as Trump detractors were responsive to the content of the message in terms of agreeing with the financial aspects of the name change, although being unaffected by the source of the message.

### 7.1.2. Moral Agreement

The main effect of Trump support was significant, indicating that Trump detractors ($M = 5.01$, $SD = 1.37$) expressed stronger moral agreement with the name change compared to Trump supporters ($M = 3.44$, $SD = 1.50$), $F(1, 243) = 72.76$, $\eta p^2 = 0.23$, $p < 0.001$. There were no other significant main or interaction effects. These results would seem to suggest that the effects of attributing a statement to Trump on Trump supporters do not carry over to moral attitudes when he is speaking strictly about the financial aspects of a topic.

### 7.2. *Exploratory PROCESS Analysis*

An exploratory moderation analysis using PROCESS Model 1 [42] demonstrated that the Source × Trump Support interaction for the financial agreement variable remained significant when Trump support was entered into the model as a continuous variable (rather than as a dichotomous variable, as in the ANOVA above): $b = 0.29$, 95% $CI$ [0.07, 0.51], $p = 0.010$. Probing this interaction demonstrated that the effect of the source manipulation on agreement ratings was significant at the 70th percentile of Trump support and higher, as illustrated in the Johnson–Neyman plot in Figure 4.

The correlation between support specifically for Trump and political orientation was once again very high, $r = 0.769$, $p < 0.001$. The results of another PROCESS Model 1 analysis with political orientation as the moderator again showed a significant interaction, $b = 0.23$, 95% $CI$ [0.01, 0.45], $p = 0.041$. Probing this interaction demonstrated that the effect of the source manipulation on agreement ratings was significant at the 77th percentile of political orientation and higher. These exploratory analyses lend additional support to the asymmetric bias hypothesis.

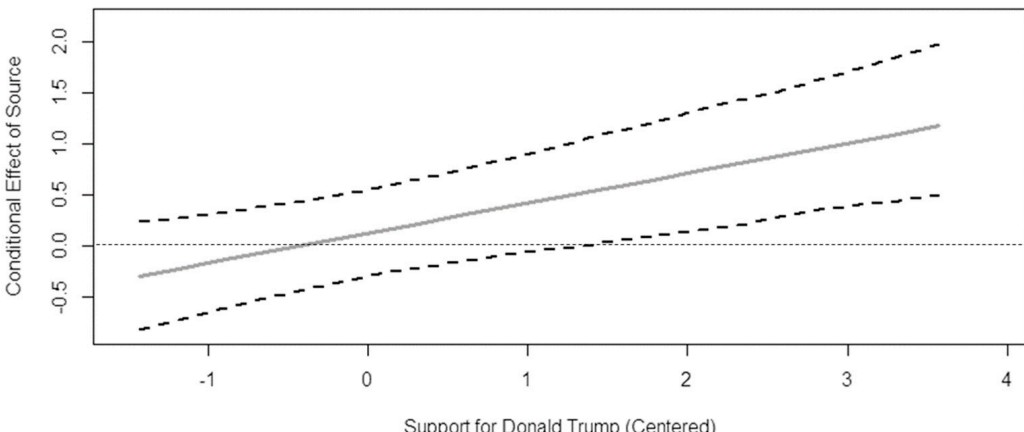

**Figure 4.** Effect of source on name change support (financial) by the degree of Trump support with 95% confidence intervals.

## 8. Discussion

The results of Study 2 once again supported the asymmetric bias hypothesis. Trump's supporters showed a tendency to agree with Donald Trump rather than an anonymous sports broadcaster, regardless of the content of the statement. Trump's effect on his supporters' differences in support for the Washington Football Team's name change only applied to financial aspects and not to moral aspects (which were not mentioned in any of the statements). Once again, exploratory analyses showed that the effects became stronger as support for Trump increased and as general conservatism increased. Trump's detractors (and conservatives) were once again not affected by the source of information. They were not reflexively disagreeing with Trump; thus, the bias was asymmetrical.

## 9. Study 3

Studies 1 and 2 were able to demonstrate an asymmetric bias whereby Trump's supporters were more likely to agree with Trump's positions regardless of their content (in comparison to the same positions espoused by anonymous sources) and Trump's detractors were not affected by whether the source of these messages was Trump or anonymous on two very different issues. However, although the previous two studies were able to reject a reflexive bias against Trump by his detractors, Studies 1 and 2 were not able to examine whether Trump's detractors would show a partisan bias in favor of the interest of Trump's Democratic rivals. Fortunately, by the end of Study 2 data collection, Trump was in the last weeks of his re-election contest with Democratic rival Joe Biden, and the days after the election provided a unique opportunity to seek symmetrical partisan bias in the context of ballot counting in so-called "swing states" or "battleground states", where candidates spend the vast majority of campaign resources, the vote totals are close, and the election is ultimately decided [45]—also tying directly into the Electoral College issue from Study 1. The COVID-19 pandemic drove record-breaking early and mail-in voting turnout [46], and the fact that many states did not allow the counting of such ballots until Election Day caused delays in several states declaring who won their Electoral College votes. Additionally, such early ballots were more likely to come from Democratic voters who took the threat of the pandemic more seriously and anticipated long lines and crowded polling places in their more heavily populated areas. Accordingly, votes for Trump were more likely to be tallied on Election Night, whereas Biden "caught up" as early and mail-in ballots were counted in subsequent days. The one swing state to buck this trend was Arizona, where Biden was up in Election Night counts and Trump seemed as though he may gain in counting throughout the week. Accordingly, we selected Arizona and one swing state that experienced a "blue shift" toward Biden over the week (Pennsylvania), randomly assigned participants to consider the ongoing vote counts in one of the two states (while reminding them of who led on Election Night and who was likely to benefit from counting remaining

ballots), and measured both Trump voters' and Biden voters' support for counting all votes in that state. If partisan bias in this context is symmetric, then Biden voters should be more likely to support continued counting in Pennsylvania than in Arizona and Trump voters should be more likely to support continued counting in Arizona than in Pennsylvania. If partisan bias in this context is asymmetric, as in Studies 1 and 2, then this self-serving pattern should only be apparent in Trump voters.

## 10. Methods

### 10.1. Participants

Using Amazon's Mechanical Turk (mTurk) system and limiting enrollment in the study to the United States, 423 participants ($M_{age}$ = 39.37, *SD* = 12.51) were recruited and paid USD 0.20 for completing the study on a single day during the week immediately after Election Day 2020. G*Power 3.1 [41] recommended a sample size of 389 to find small effects (*f* = 0.15) with power = 0.80 at conventional levels of significance (*p* < 0.05) for the planned analysis of variance. Accordingly, data collection was halted shortly after, providing leeway for the possibility of data exclusions for failing attention checks. The sample was 52% female, 72% non-Hispanic white, had a median education of a bachelor's degree, and a median income between USD 50,000 and USD 74,999. The sample was more strongly some denomination of Christian (48%) than non-religious (33%). It should be noted that to allow for data collection within a few hours (to avoid the possibility of major changes in the status of the election), this was the only study in this manuscript not to limit data collection to mTurk "Master Workers".

### 10.2. Materials and Procedure

After agreeing to complete the survey on mTurk, participants followed a link to a survey hosted on Qualtrics where they completed an online consent document. Participants read one of two passages on the ongoing counting of ballots in either Arizona or Pennsylvania. Participants in both conditions were reminded who the count favored on Election Night (Biden in Arizona; Trump in Pennsylvania) and which candidate was predicted to benefit from counting all remaining votes (Trump in Arizona; Biden in Pennsylvania). See Appendix C for these materials. As the dependent measure, participants indicated their level of support for "counting all the votes" in that particular state on 7-point scales, with high values indicating more support. Afterwards, participants answered two attention/manipulation check items asking them to identify which state they were asked about and which candidate could benefit in that state from counting all votes. Only those who passed both checks were included in analyses. Participants also indicated who they voted for: Biden, Trump, or "someone else/did not vote." Only participants indicating that they voted for Biden or Trump were included in analyses. These two exclusion criteria reduced the sample size by 37 (n = 386). Participants also reported their political orientation on a 1 to 7 scale, their perceptions of Trump's job performance on a 1 to 7 scale, completed a four-item measure of racial bias commonly used in the American National Election Survey, and completed a demographic questionnaire, received a completion code, and returned to mTurk to enter the code.

## 11. Results

### 11.1. Planned Analyses

A 2 (2020 vote: Biden vs. Trump) × 2 (target state: Arizona vs. Pennsylvania) analysis of variance was conducted, with the level of support for counting all votes in the target state as the outcome variable. The main effect of the 2020 vote was significant, indicating overall higher levels of support for counting all votes from Biden voters (*M* = 6.45, *SD* = 1.15) than Trump voters (*M* = 5.95, *SD* = 1.68), $F(1, 382)$ = 13.49, $\eta p^2$ = 0.034, *p* < 0.001. The main effect of the target state was significant as well, indicating greater support for counting all ballots in Arizona (*M* = 6.40, *SD* = 1.20) than in Pennsylvania (*M* = 6.08, *SD* = 1.58), $F(1, 382)$ = 9.00, $\eta p^2$ = 0.023, *p* = 0.003.

Both significant main effects were qualified by a Vote 2020 × Target State interaction, $F(1, 382) = 14.00$, $\eta p^2 = 0.035$, $p < 0.001$. Probing this interaction revealed that Trump voters were more likely to support counting all votes in Arizona ($M = 6.41$, $SD = 1.18$), where it could help Trump, rather than in Pennsylvania ($M = 5.46$, $SD = 1.98$), *95% CI$_{Diff}$* [0.53, 1.37], $p < 0.001$, whereas Biden voters were not significantly affected by the target state, $M_{AZ} = 6.40$, $SD_{AZ} = 1.23$, $M_{PA} = 6.50$, $SD_{PA} = 1.06$, *95% CI$_{Diff}$* [−0.46, 0.25], $p = 0.515$. This pattern is consistent with Hypothesis 1 (Asymmetric Trump Bias) and inconsistent with Hypothesis 2 (Symmetric Trump Bias), and is also inconsistent with a similar bias among Trump's detractors in favor of the interests of Democrats (see Figure 5).

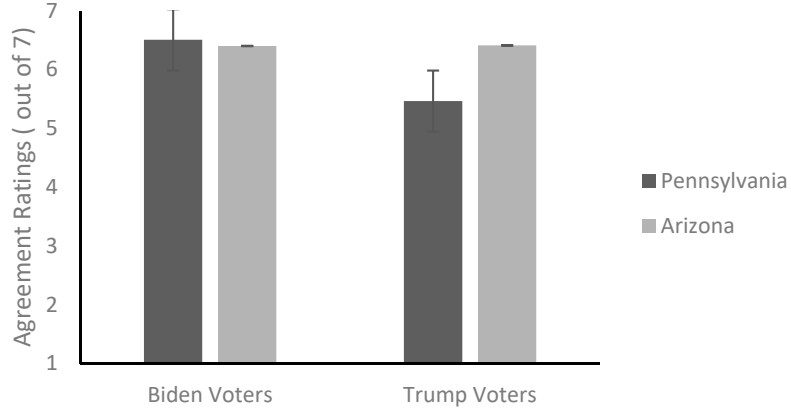

**Figure 5.** Support for counting all votes in the 2020 election with target states.

### 11.2. Exploratory PROCESS Analysis

Similarly to Studies 1 and 2, Study 3 found a very strong positive correlation between the continuous measure of Trump support (this time in perceptions of his job performance) and general political orientation, $r = 0.721$, $p < 0.001$. The inclusion of a measure of racial bias also allowed us to show that racial bias was highly and positively related to both Trump support, $r = 0.608$, $p < 0.001$, and political orientation (conservatism), $r = 0.627$, $p < 0.001$, which is consistent with prior research (e.g., Schaffner et al., 2018).

Accordingly, an exploratory moderation analysis was performed using PROCESS Model 6 [42], which allows for tests of serial mediation. In order to show a possible series of psychological mechanisms resulting in Trump supporters having, on average, a higher endorsement of subverting democracy when it suits their needs that could be generalized to other contexts where racial and political attitudes are relevant, we examined the effects of racial bias (x) on political orientation (m1), then subsequently on Trump support (m2), and ultimately the willingness to stop the vote for Trump's benefit in Pennsylvania (y). See Figure 6 for an illustration of the direct and serial indirect pathways. We sequenced the variables in this order because it is likely that, developmentally speaking, racial biases develop prior to broad political orientations, which develop prior to specific political attitudes. Indirect effects were estimated for each of 5000 bootstrapped samples and were assumed to be significant if the 95% confidence intervals did not contain zero. Indirect effects did not produce exact *p*-values. Results of the serial mediation analysis showed that the indirect effect of racism through the mediators of general political orientation and Trump support was indeed significant, $b = 0.050$, *95% CI* [−0.731, −0.283], indicating a possible pathway of mechanisms whereby increased racism motivates general political conservatism, which motivates Trump support, and ultimately the willingness to disenfranchise votes to Trump's benefit.

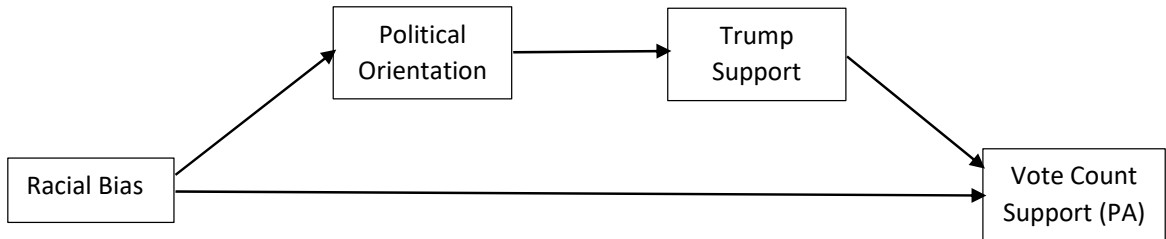

**Figure 6.** Conceptual serial mediation model.

## 12. Discussion

Study 3, conducted in the midst of a bitterly contested election, once again supported an asymmetric bias effect whereby Trump voters were more likely to agree that votes should be counted when it may benefit him (but not when it may benefit his opponent), whereas Biden voters strongly agreed that all votes should be counted regardless. It should be noted that a majority of both Biden voters and Trump voters indicated the strongest possible support for counting all votes regardless of which candidate would benefit, although nearly half of Trump voters were willing to subvert democracy to some degree in order to benefit Trump. This is consistent with the bizarre simultaneous demonstrations by Trump supporters in Arizona—where they chanted "Count those votes!"—and Michigan—where the situation was similar to Pennsylvania, and Trump supporters chanted "Stop the count!".

## 13. General Discussion

Across all studies, we found an asymmetrical change in preferences in response to the ostensible positions of Donald Trump: Trump's supporters consistently shifted their attitudes to more closely match ostensible opinions and the real-life interests of Trump whereas even Trump's strongest detractors did not reflexively oppose Trump. In Studies 1 and 2, the results were the same regardless of whether the statements made by Trump were politically congenial to more liberal or more conservative positions. This asymmetry in responding to the source of information may be explained by cognitive asymmetries between liberals and conservatives found in prior research, such as differences in thinking style [26], cognitive ability [22], and the acceptance of "bullshit" [47,48]. Results of the current study do not support the broad existence of so-called "Trump Derangement Syndrome" on the left, but they may lend credence to accusations that some Trump supporters have a cult-like loyalty to the 45th president. Trump's detractors, far from reflexively rejecting any claims he makes, may instead generally ignore information from the 45th president and not allow it to affect their views one way or another. However, this does not preclude Trump affecting attitudes among his detractors in other ways [49].

### 13.1. Implications

The implications of these results are apparent in the responses of Trump voters in the days immediately following the 2020 U.S. Presidential Election as they demonstrated outside vote-counting locations in Arizona chanting "Count those Votes!", and in Michigan chanting "Stop the Count!" Meanwhile, Biden supporters consistently called for all votes to be counted regardless. Only one side had a substantial contingent of individuals perfectly willing to be transparently self-serving. Republican elected officials followed suit by denying Biden's clear victory; these actions followed years of conspiracy theories and assaults on democracy [50], such as voter suppression, gerrymandering, and supporting the inequities of the Electoral College, and ultimately led to attempted insurrection.

The fact that asymmetrical biases are apparent in broad political behavior and in myriad research findings makes the academic debate on the issue seem almost as unbalanced. Baron and Jost [31] spelled out some reasons why researchers may be compelled to interpret their research findings as supporting the notion that political bias is symmetrical. Motivation to appear unbiased may be elicited by calls from prominent researchers to

promote "ideological diversity" in social psychology [51–53]. It may also be motivated by a conscious or unconscious recognition of the fact that there is a higher rejection rate for manuscripts on "liberal" topics [54]. Both the ideological diversity movement and the relative reticence to publish results that are amenable to liberal political positions may stem from a desire to make the field seem "unbiased" and improve its reputation among a distrustful general public.

The results of these studies should also raise concern about the dangers Trump may pose even as a private citizen. Citizen Trump still has millions of devoted followers, and he may influence them to continue to deny reality and subvert democracy, perhaps even violently, as demonstrated during the assault on the U.S. Capitol Building on 6 January 2021. Broadly speaking, these results speak to the clear and present danger posed by other extremist leaders similar to Trump in countries around the world: that their violent rhetoric and spreading of misinformation will foment violence and destruction from their impressionable followers. In addition, racial bias is a strong predictor of Trump support [55], and similar far-right leaders may also exploit racial animus to weaken the social safety net in other nations facing ongoing immigration crises [56].

*13.2. Limitations and Future Directions*

Studies 1 and 2 were limited due to lacking a source that liberals find favorable (no statements were attributed to Bernie Sanders, for example). Study 3 provided the opportunity for liberals to demonstrate bias in favor of "their team" rather than just bias against Trump. It is possible that Biden supporters were confident that the Arizona vote count would turn out in Biden's favor regardless. Future research could improve upon this by ensuring that liberals have an opportunity to show favoritism toward their own in a situation where they may have less confidence that remaining unbiased would not result in negative consequences. However, it should be noted that typical suggestions for doing so are also problematic. There really is no liberal equivalent to Donald Trump in the United States that could have been used as a counterpoint in this study. Prominent Democrats such as Joe Biden and Hillary Clinton are centrists. It would be a false equivalence to consider the United States' most prominent progressive politician, Bernie Sanders, as being equally as extreme to the left as Trump is to the right. Additionally, although Sanders has a devoted following, it is a following based on his decades of unwavering consistency across a range of policy issues, and it is unlikely that researchers could fool Sanders's staunchest supporters into thinking he advocated for positions that in reality he does not. The current group of studies, particularly with the opportunity seized for Study 3, represent, in our minds, close to the best first step possible in examining symmetries and asymmetries in bias regarding Donald Trump, which may be generalizable to other similar leaders and their ability to exploit racial bias that feeds extreme, authoritarian conservatism.

Future research could also examine moderation and mediation effects involving attitudes, cognitions, and feelings. One such area to seek moderating influences or mediating mechanism is within the realm of Moral Foundations Theory [57]. Moral foundations have been demonstrated to predict numerous politically relevant outcomes [58–60].

Additional research could be conducted to test for (a)symmetric biases and behavioral responses to emotional experiences in other opposing groups such as atheists and Christians in the United States [61–63], as well as stigma-by-association effects [64] induced by supporting certain leaders. Finally, the exploratory findings from Study 3 regarding the relationship between increased anti-Black racial bias and support for the subversion of democracy should also generate further research [65].

## 14. Conclusions

The fact that political bias is not symmetrical in the United States should be apparent not only from prior research on the psychological differences between liberals and conservatives, but also from political realities taking place at the time of this writing. Trump supporters in the current research supported Trump's ostensible interests and positions

regardless of what those interests and positions were, a finding that could explain the attempted January 6th insurrection and other attempts to undermine the results of the 2020 election in Trump's favor, such as the biased 'Cyber Ninja' ballot audit in Arizona. Meanwhile, the American left is much faster to lose support for Democratic leaders such as Joe Biden [66]. In light of such results and real-world events, researchers and publishers cannot be so fearful of appearing biased ourselves that we contribute to the prevailing narrative of false equivalence.

**Author Contributions:** Conceptualization, A.S.F. and F.H.; methodology, A.S.F. and F.H.; formal analysis, A.S.F.; data curation, A.S.F.; writing—original draft preparation, A.S.F. and F.H.; writing—review and editing, A.S.F. and F.H.; visualization, F.H.; supervision, A.S.F.; out-of-pocket funding expenditures, A.S.F. and F.H. All authors have read and agreed to the published version of the manuscript.

**Funding:** This research received no external funding.

**Institutional Review Board Statement:** The study was conducted according to the guidelines of the Declaration of Helsinki, and approved by the Institutional Review Board of Central Michigan University (2019-780; 2019-1120; 2020-397).

**Informed Consent Statement:** Informed consent was obtained from all subjects involved in the study.

**Data Availability Statement:** Data, hypotheses, and materials are available at https://osf.io/wt635/ (accessed on 3 September 2021).

**Conflicts of Interest:** The authors declare no conflict of interest.

## Appendix A

Statements on National Popular Votes from Anonymous Source*

"Trump won the electoral college, but he lost the popular vote. Losing the popular vote doesn't matter. He was not trying to win a popular vote. I have always favored the Electoral College. And the fact that Hillary Clinton would be our president if we had a popular vote just cements my support for the Electoral College. I want to maintain the current Electoral College system. Trump did not need to go to New York or California to campaign because he did not need to win those states to win the Electoral College. If the election were based on total popular vote, he would have had to appeal to liberals in New York and California, and he still would not have stood a chance of winning the popular vote. Switching from the Electoral College to a popular vote would permanently hand the presidency over to the Democrats, so I do not want to see a national popular vote."

"Trump won the electoral college, but he lost the popular vote. It doesn't matter. He was not trying to win a popular vote. It is possible he could win the popular vote if he was trying to do that, so I don't think it matters to me which one we use. With the Electoral College, both candidates focus on winning a few swing states because most states are solidly Republican or Democrat. If the election were based on total popular vote, Trump would have gone to New York and California, and it's possible he may have won a lot more votes in those states by campaigning there. Switching from the Electoral College to a popular vote is a non-issue as far as I am concerned."

"Trump won the electoral college, but he lost the popular vote. It doesn't matter. He was not trying to win a popular vote. I have always favored the popular vote, and the results of this election don't change my mind. I would rather see it where you went with a national popular vote. Trump didn't campaign in New York or California because he did not need to win those states to win the Electoral College. If the election were based on total popular vote, he would have campaigned in New York and California and won a lot more votes in those states and he would have won the popular vote by a very large margin. He would

also help the prospects of Republicans running for other offices in those states by increasing turnout. I like the idea of campaigning all over the country to win as many votes as possible, so I still want to see a national popular vote."

*Trump versions of statements were written in first-person.

**Appendix B**

Statements on Washington Football Team name change (identical regardless of source).

"The name change was absolutely a mistake from a financial standpoint. Giving in to the politically correct minority will alienate fans of the team and fans of the league in general. In the long-term, the Washington Football Team will sell less merchandise and fewer tickets than the Washington Redskins, and many fans will stop watching the NFL due to the increasing focus on politics."

"The name change was absolutely the right call from a financial standpoint. It was long overdue to avoid losing fans and losing money. The Washington Football Team will sell more merchandise now that fans need new hats, shirts, and jerseys to replace their old Redskins stuff. Then they can sell those same fans new merchandise again when they finally pick their permanent replacement name."

**Appendix C**

Ballot Counting Statements.

"On Election Night, Joe Biden led in the Presidential vote count in the US State of Arizona. However, he was not yet officially declared the winner of the state because there were enough uncounted ballots that it was possible Donald Trump could win it instead."

"On Election night, Donald Trump led in the Presidential vote count in the US State of Pennsylvania. However, he was not yet officially declared the winner of the state because there were enough uncounted ballots that it was possible Joe Biden could win it instead."

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
