# Peer review of "Seeking Evidence of The MAGA Cult and Trump Derangement Syndrome: An Examination of (A)symmetric Political Bias"

_societies, doi:10.3390/soc11030113_

Round 1

Reviewer 1 Report

The reviewed paper is settled in the context of social and political psychology. The focus is on Donald Trump and the behaviors and perceptions of his political supporters and opponents. The authors of this paper explore the existence of (a)symmetric bias regarding Donald Trump: Do Trump supporters consistently bias all messages and statements in favor of Trump’s political interests? Is the opposite true for Trump’s detractors? The authors present the results of three empirical studies that build on each other. All studies are web-based survey studies with a sufficient sample size (n=394, n=251, and n=423, for study 1, study 2, and study 3, respectively). Statistical methods used in this paper are descriptive statistics, analyses of variances (ANOVAs), exploratory moderation analyses, and serial mediation analyses. The authors discuss the results of all three studies in separate discussion chapters. In addition, they present practical implications and limitations of this article in a general discussion section.

I really enjoyed reading this article and thank the authors for the opportunity to review their interesting paper. In my opinion, this paper deals with a very important and timely topic, especially in times of conspiracy theories, political “cults” (e.g., QAnon), and so-called “fake news”. The paper is well-written and structured. The introduction guides the reader smoothly into this topic, all necessary background information (i.e., citing appropriate literature) is given. It was a pleasure to read these sections!

Presented empirical results are shocking, mainly the interaction effects: biased perceptions and behaviors were found in the group of Trump supporters only. It is like Trump is the “puppet master” controlling his supporters. Indeed, the authors wrote: “Trump’s followers are often derided as a cult [12] blindly accepting the words of their authoritarian leader as gospel […]” (page 2, line 41).

As a statistician, I very liked how the authors used their statistical methods. Everything was correct and all steps of data analyses were clearly described. For all three studies, the authors included a sample size calculation mentioning (almost) all necessary parameters. Results of ANOVAs were presented transparently (i.e., mean, standard deviation, F-value, effect size, and p-value). Additional analyses (e.g., serial mediation analyses in study 3, page 11, line 429), were useful in understanding how observed effects (e.g., Trump support) develop in human minds.

While reading this paper I detected only some small errors and miss-spellings. Hence, I have some comments/suggestions that I hope will help the authors to further develop this line of work:

  1. Introduction (page 2, line 60): not “[…] among conservatives [14]Conservatives have been […]” but rather “[…] among conservatives [14]. Conservatives have been […]”. Please add a space and a dot in this sentence.
  2. Introduction (page 2, line 72): not “[…] found to contribute to contribute to polarization […]” but rather “[…] found to contribute to polarization […]”. Please delete the second “to contribute” in this sentence.
  3. Sample size calculation of study 1 (page 3, line 136), study 2 (page 6, line 249), and study 3 (page 10, line 381): Please add which statistical test you used in this sample size calculation (probably ANOVA) and cite G*Power appropriately: Faul, F., Erdfelder, E., Lang, A.-G., & Buchner, A. (2007). G*Power 3: A flexible statistical power analysis program for the social, behavioral, and biomedical sciences. Behavior Research Methods, 39, 175-191. Faul, F., Erdfelder, E., Buchner, A., & Lang, A.-G. (2009). Statistical power analyses using G*Power 3.1: Tests for correlation and regression analyses. Behavior Research Methods, 41, 1149-1160.
  4. Results of study 1 (page 4, line 182 and 184): p-values are missing in these two sentences (e.g., “(M = 3.82, SD = 1.81), 95% CIDiff [-1.32, -0.31]”, page 4, line 182). Please add p-values to these sentences as you did this correctly in the next paragraph (e.g., “(M = 2.87, SD = 1.54), 95% CIDiff [0.83, 1.94], p < .001”, page 5, line 189).
  5. Results of study 1 (page 5, line 190): Please do not present results which are “marginally” significant (“(p=.06)”, page 5, line 190). The error rate on the 10% significance level is obviously too high.
  6. Results of study 2 (page 7, line 288): Please delete the unnecessary space in this sentence (“[…] ps > .06.   The three-way interaction was also 288 non-significant.”, page 7, line 288).
  7. Results of study 2 (page 7, line 295, page 8 line 298, page 8 line 305): As mentioned above, p-values are missing in these sentences. Please add p-values for these disassembled interaction effects (compare this to the correct presentation on page 5, line 189).
  8. Results of study 2 (page 8, line 310): not “[…] while being unaffected the message source.” but rather “while being unaffected by the message source.” Please correct this sentence.
  9. Results of study 2 (page 9, line 337): Something strange happened to font size. This paragraph seems to be too small.
  10. Results of study 3 (page 11, line 418): Please use a consistent writing style and use the same variable label throughout the whole manuscript: Please compare the variable label “2020 vote” (page 11, line 411) to “Vote 2020” (page 11, line 418).
  11. Results of study 3 (page 11, line 421, and page 11 line 423): As mentioned above, p-values are missing in these sentences. Please add p-values for these disassembled interaction effects (compare this to the correct presentation on page 5, line 189).
  12. General Discussion (page 13, line 510): Please use a consistent writing style throughout the whole manuscript: Please compare “Studies 1 and 2” (page 12, line 470) to “Studies 1 & 2” (page 13, line 510).

Author Response

The reviewed paper is settled in the context of social and political psychology. The focus is on Donald Trump and the behaviors and perceptions of his political supporters and opponents. The authors of this paper explore the existence of (a)symmetric bias regarding Donald Trump: Do Trump supporters consistently bias all messages and statements in favor of Trump’s political interests? Is the opposite true for Trump’s detractors? The authors present the results of three empirical studies that build on each other. All studies are web-based survey studies with a sufficient sample size (n=394, n=251, and n=423, for study 1, study 2, and study 3, respectively). Statistical methods used in this paper are descriptive statistics, analyses of variances (ANOVAs), exploratory moderation analyses, and serial mediation analyses. The authors discuss the results of all three studies in separate discussion chapters. In addition, they present practical implications and limitations of this article in a general discussion section.

I really enjoyed reading this article and thank the authors for the opportunity to review their interesting paper. In my opinion, this paper deals with a very important and timely topic, especially in times of conspiracy theories, political “cults” (e.g., QAnon), and so-called “fake news”. The paper is well-written and structured. The introduction guides the reader smoothly into this topic, all necessary background information (i.e., citing appropriate literature) is given. It was a pleasure to read these sections!

Presented empirical results are shocking, mainly the interaction effects: biased perceptions and behaviors were found in the group of Trump supporters only. It is like Trump is the “puppet master” controlling his supporters. Indeed, the authors wrote: “Trump’s followers are often derided as a cult [12] blindly accepting the words of their authoritarian leader as gospel […]” (page 2, line 41).

As a statistician, I very liked how the authors used their statistical methods. Everything was correct and all steps of data analyses were clearly described. For all three studies, the authors included a sample size calculation mentioning (almost) all necessary parameters. Results of ANOVAs were presented transparently (i.e., mean, standard deviation, F-value, effect size, and p-value). Additional analyses (e.g., serial mediation analyses in study 3, page 11, line 429), were useful in understanding how observed effects (e.g., Trump support) develop in human minds.

Thank you for the thorough reading and positive feedback on our work!

While reading this paper I detected only some small errors and miss-spellings. Hence, I have some comments/suggestions that I hope will help the authors to further develop this line of work:

  1. Introduction (page 2, line 60): not “[…] among conservatives [14]Conservatives have been […]” but rather “[…] among conservatives [14]. Conservatives have been […]”. Please add a space and a dot in this sentence. Thanks for noticing this typing error. It has now been fixed.
  2. Introduction (page 2, line 72): not “[…] found to contribute to contribute to polarization […]” but rather “[…] found to contribute to polarization […]”. Please delete the second “to contribute” in this sentence. Thanks for noticing this typing error. It has now been fixed.
  3. Sample size calculation of study 1 (page 3, line 136), study 2 (page 6, line 249), and study 3 (page 10, line 381): Please add which statistical test you used in this sample size calculation (probably ANOVA) and cite G*Power appropriately: Faul, F., Erdfelder, E., Lang, A.-G., & Buchner, A. (2007). G*Power 3: A flexible statistical power analysis program for the social, behavioral, and biomedical sciences. Behavior Research Methods, 39, 175-191. Faul, F., Erdfelder, E., Buchner, A., & Lang, A.-G. (2009). Statistical power analyses using G*Power 3.1: Tests for correlation and regression analyses. Behavior Research Methods, 41, 1149-1160. Thanks for providing the full reference. The suggested changes have been made. We referenced the latter of the two as version 3.1 was used.
  4. Results of study 1 (page 4, line 182 and 184): p-values are missing in these two sentences (e.g., “(M = 3.82, SD = 1.81), 95% CIDiff [-1.32, -0.31]”, page 4, line 182). Please add p-values to these sentences as you did this correctly in the next paragraph (e.g., “(M = 2.87, SD = 1.54), 95% CIDiff [0.83, 1.94], p < .001”, page 5, line 189). Missing p values have now been added.
  5. Results of study 1 (page 5, line 190): Please do not present results which are “marginally” significant (“(p=.06)”, page 5, line 190). The error rate on the 10% significance level is obviously too high. Agreed. The sentence now reads "For Trump detractors this difference between the anti-NPV and neutral statements was nonsignificant (p = .06). "
  6. Results of study 2 (page 7, line 288): Please delete the unnecessary space in this sentence (“[…] ps > .06.   The three-way interaction was also 288 non-significant.”, page 7, line 288). Thanks for noticing this typing error. It has now been fixed.
  7. Results of study 2 (page 7, line 295, page 8 line 298, page 8 line 305): As mentioned above, p-values are missing in these sentences. Please add p-values for these disassembled interaction effects (compare this to the correct presentation on page 5, line 189). Missing p values have now been added.
  8. Results of study 2 (page 8, line 310): not “[…] while being unaffected the message source.” but rather “while being unaffected by the message source.” Please correct this sentence. The sentence has been rewritten as suggested. Thank you!
  9. Results of study 2 (page 9, line 337): Something strange happened to font size. This paragraph seems to be too small.
  10. Results of study 3 (page 11, line 418): Please use a consistent writing style and use the same variable label throughout the whole manuscript: Please compare the variable label “2020 vote” (page 11, line 411) to “Vote 2020” (page 11, line 418). Good formatting catch. Thanks! The font size has been increased from 9 to 10 pt to be consistent with the rest of the paper.
  11. Results of study 3 (page 11, line 421, and page 11 line 423): As mentioned above, p-values are missing in these sentences. Please add p-values for these disassembled interaction effects (compare this to the correct presentation on page 5, line 189). Missing p values have now been added.
  12. General Discussion (page 13, line 510): Please use a consistent writing style throughout the whole manuscript: Please compare “Studies 1 and 2” (page 12, line 470) to “Studies 1 & 2” (page 13, line 510). The ampersand '&' in the second instance has been changed to the word 'and' for consistency. Thanks again for your careful eye!

Reviewer 2 Report

Thank you for this opportunity to revise the manuscript titled "Seeking Evidence of The MAGA Cult and Trump Derangement Syndrome: An Examination of (A)symmetric Political Bias " that was submitted to Societies.
The mentioned work is timely and will certainly interest the readers.
In addition there are some comments about the manuscript which are listed below:
Firstly, it would be good if the authors had given an explanation of where such a state of mind of American conservatives came from. However, I understand that this is just an article. And the answer to this question would be a monograph.
Secondly, I would suggest that the author (s) should delete the sentence, which contains a hint of neurocognitive differences between liberals and conservatives (lines 81 and 82). Attempts to explain socio-psychological phenomena by differences in physiology are doubtful.
I am sure that the answers to these questions /comments  will improve the quality of this paper. I will be happy to review the revised manuscript.

Author Response

Thank you for this opportunity to revise the manuscript titled "Seeking Evidence of The MAGA Cult and Trump Derangement Syndrome: An Examination of (A)symmetric Political Bias " that was submitted to Societies.
The mentioned work is timely and will certainly interest the readers.
In addition there are some comments about the manuscript which are listed below:

Thanks for the compliments on our work!

Firstly, it would be good if the authors had given an explanation of where such a state of mind of American conservatives came from. However, I understand that this is just an article. And the answer to this question would be a monograph.

Secondly, I would suggest that the author (s) should delete the sentence, which contains a hint of neurocognitive differences between liberals and conservatives (lines 81 and 82). Attempts to explain socio-psychological phenomena by differences in physiology are doubtful.

We think that the discussion of structural differences provides a potential partial answer to your previous comment. Some of the structural differences are in areas of the brain that evidence has shown to be heavily involved in intolerance of ambiguity (ACC) and negative emotional responses (amygdala). The behaviors associated with such differences may feed into asymmetries in political biases. Similar information has now been added to the manuscript.

I am sure that the answers to these questions /comments  will improve the quality of this paper. I will be happy to review the revised manuscript.